# Site-Specific Conversion of Cysteine in a Protein to Dehydroalanine Using 2-Nitro-5-thiocyanatobenzoic Acid

**DOI:** 10.3390/molecules26092619

**Published:** 2021-04-29

**Authors:** Yuchen Qiao, Ge Yu, Sunshine Z. Leeuwon, Wenshe Ray Liu

**Affiliations:** 1The Texas A&M Drug Discovery Laboratory, Department of Chemistry, Texas A&M University, College Station, TX 77843, USA; richyqiao@tamu.edu (Y.Q.); gyu@tamu.edu (G.Y.); sunshineleeuwon@gmail.com (S.Z.L.); 2Department of Biochemistry & Biophysics, Texas A&M University, College Station, TX 77843, USA; 3Molecular & Cellular Medicine Department, College of Medicine, Texas A&M University, College Station, TX 77843, USA; 4Institute of Biosciences and Technology and Department of Translational Medical Sciences, College of Medicine, Texas A&M University, Houston, TX 77030, USA

**Keywords:** cysteine, dehydroalanine, 2-nitro-5-thiocyanatobenzoic acid, NTCB, post-translational modifications

## Abstract

Dehydroalanine exists natively in certain proteins and can also be chemically made from the protein cysteine. As a strong Michael acceptor, dehydroalanine in proteins has been explored to undergo reactions with different thiolate reagents for making close analogues of post-translational modifications (PTMs), including a variety of lysine PTMs. The chemical reagent 2-nitro-5-thiocyanatobenzoic acid (NTCB) selectively modifies cysteine to form *S*-cyano-cysteine, in which the S–Cβ bond is highly polarized. We explored the labile nature of this bond for triggering E2 elimination to generate dehydroalanine. Our results indicated that when cysteine is at the flexible *C*-terminal end of a protein, the dehydroalanine formation is highly effective. We produced ubiquitin and ubiquitin-like proteins with a *C*-terminal dehydroalanine residue with high yields. When cysteine is located at an internal region of a protein, the efficiency of the reaction varies with mainly hydrolysis products observed. Dehydroalanine in proteins such as ubiquitin and ubiquitin-like proteins can serve as probes for studying pathways involving ubiquitin and ubiquitin-like proteins and it is also a starting point to generate proteins with many PTM analogues; therefore, we believe that this NTCB-triggered dehydroalanine formation method will find broad applications in studying ubiquitin and ubiquitin-like protein pathways and the functional annotation of many PTMs in proteins such as histones.

## 1. Introduction

The currently known genetic code specifies 22 proteinogenic amino acids that are biosynthetically incorporated into proteins during translation. Built upon them, organisms, especially eukaryotes, undergo a plethora of post-translational modifications (PTMs) to impart proteins with a large variety of unique functions. Functional annotation of PTMs requires the synthesis of proteins that contain them. In cells, proteins are usually post-translationally modified via enzymatic reactions. Due to the fact that the enzymes which install PTMs are either unknown or promiscuous, chemical biologists have long been focusing on the development of synthetic and semi-synthetic methods for the synthesis of post-translationally modified proteins for their functional investigations [1,2,3,4]. Notable methods include native chemical ligation [5], expressed protein ligation [6], and the genetic code expansion technique that relies on the suppression of amber codon for the incorporation of noncanonical amino acids that are post-translationally modified proteinogenic amino acids themselves, or can be chemically modified to form amino acids with PTMs [7,8,9,10,11]. Built upon the unique reactivity of the most nucleophilic amino acid, cysteine, methods have also been developed for its conversion to a variety of PTM analogues via both direct conjugation to many electrophiles [12] and chemical conversion to dehydroalanine (Dha) [13]. Dha contains an α, β-unsaturated carbonyl moiety that can undergo both Michael addition and cross coupling reactions to generate PTMs or their analogues [14,15]. Related applications include the functional annotation of histone PTMs [16]. Moreover, Dha in a protein might form a reactive chemical probe for covalent conjugation with associating partners for their identification. Various mono-/poly-ubiquitin (Ub) or ubiquitin-like protein (Ubl)-based Dha probes have been successfully used to capture enzymes functioning in Ub and Ubl pathways [17,18,19,20]. Those Ub/Ubl-Dha molecules have also been used for the synthesis of ubiquitinated or Ubl-tagged proteins to discover the selectivity and linkage specificity of deubiquitinases (DUBs) and ubiquitin-like proteases (ULPs) [21,22,23].

To chemically convert cysteine to Dha in a protein in relatively mild conditions, Davis et al. developed several reagents [14]. Two of them, 2,5-dibromohexanediamide (DBHDA) [14] and methyl 2,5-dibromopentanoate (MDBP) [24], are the most commonly used which selectively convert cysteine to Dha by bis-alkylation and then elimination in the aqueous system (Scheme 1A). Both reagents have been successfully practiced in the synthesis of antibody–drug conjugates and proteins with PTM analogues [14,24,25]. For the efficient conversion of cysteine to Dha when using DBHDA and MDBP, a high pH is typically used to achieve a high yield. Sometimes, a high temperature might also be required [24,25]. There are other methods available, although they generally require harsh conditions that are not compatible with proteins. Dha can also be generated in a protein through the genetic incorporation of an alkylated selenocysteine followed by oxidation and then elimination (Scheme 1B,C), or a phosphoserine followed by elimination (Scheme 1D), as demonstrated by Schultz, Liu, Park, Chen, and their coworkers [26,27,28,29]. Another technique called genetically encoded chemical conversion (GECCO), in which serine/threonine is selectively converted to Dha or dehydrobutyrine (Dhb) by the assistance of a proximal, genetically encoded fluorosulfo-tyrosine (FSY) residue, was also reported (Scheme 1E) [30]. In comparison to cysteine modification methods, these latter methods require complicated direct evolution and selection processes of t-RNA synthesases that can specifically incorporate those unnatural amino acids (uAA) into a protein. This genetic code expansion technique usually limits the expression level of those uAA-containing proteins. In this study, we report a simple method that uses the readily available, highly water-soluble 2-nitro-5-thiocyanatobenzoic acid (NTCB) to react with cysteine in a protein for its conversion to Dha under bio-orthogonal conditions at pH 7. This approach is exceedingly selective and efficient when the cysteine is located at a flexible end of a protein compared to the aforementioned methods, which makes it an ideal way to obtain several Ub- and Ubl-based, Dha-containing probes.

## 2. Results

NTCB is a highly reactive reagent that transfers its cyano group rapidly to a nucleophilic thiolate. When it is provided to a protein, it will quickly cyanylate the protein cysteine to form *S*-cyano-cysteine which undergoes reversible intramolecular addition with the cysteine *N*-amide to generate 1-acyl-2-iminothiazolidine, an intermediate that can undergo nucleophilic acyl substitution (Figure 1A). By taking advantage of this nucleophilic acyl substitution, we have recently developed a technique termed activated cysteine-directed protein ligation (ACLP) [31]. During the development of ACPL, we noticed that a protein was usually not fully converted to a ligation product. The SDS-PAGE analysis of the reaction showed a protein band that ran at the same place as the original protein (Appendix A). We previously thought that this was the unreacted original protein. HPLC analysis of products after the ACPL reaction between Ub-G76C-6H and propargylamine (Pa) showed two protein peaks (Figure 1B). Electrospray ionization mass spectrometry (ESI-MS) analysis of Peak 2 indicated the desired product from the ACPL reaction (Figure 1C). However, the deconvoluted ESI-MS spectrum of Peak 1 revealed a molecular weight of 9454.4 Da (Figure 1D) that did not match the theoretical mass of Ub-G76C-6H (9433.7 Da), disproving our original assumption of incomplete conversion of Ub-G76C-6H. This detected molecular weight matches the replacement of the cysteine thiol group in Ub-G76C-6H with propargylamine. This replacement can potentially happen in two possible pathways: the cyanylation of the cysteine thiolate makes it an easy leaving group to undergo an SN2 reaction with propargylamine or the formed thiocyanate–protein adduct undergoes a beta elimination reaction to generate a Dha residue which then reacts with propargylamine via aza-Michael addition (Figure 1E). Dha formation via the beta elimination of *S*-cyano-cysteine was previously predicted and observed at a very low level when NTCB was used to hydrolyze proteins at a high pH [32,33,34]. We deemed that this beta elimination to generate Dha and then addition with propargylamine is a more probable mechanism for the formation of the Peak 1 protein. Based on the HPLC chromatogram in Figure 1B, the Peak 1 protein is about 40% of the overall final products. This high level of Dha formation is likely due to a relatively milder pH we used to perform the ACPL reaction in comparison to that for the traditional hydrolysis reaction.

According to the scheme shown in Figure 1E, the substitution with Pa and the beta elimination are two competing reactions. Withdrawing Pa from the reaction will potentially improve the yield toward Dha and also keep Dha intact (Figure 2A). We used Ub-G76C-6H to test this prospect. After incubation in a buffer containing 0.5 mM Tris(2-carboxyethyl)phosphine (TCEP) and 5 mM NTCB at pH 9 and 37 °C overnight, about 80% of Ub-G76C-6H was successfully converted to Ub-G76Dha-6H (Figure 2B and Appendix A, Table 1), confirming an improved Dha formation yield in the absence of a strong amine nucleophile. The major byproduct was determined to be Ub_(1-75)_, a hydrolysis product in which a water molecule substituted G76C-6H (Figure 2C). We then further optimized the reaction condition by decreasing the pH value to reduce the undesirable hydrolysis. After overnight incubation, LC-MS analysis of reaction products indicated that as pH decreased, the percentage of Ub-G76Dha-6H kept increasing. The yield ranged from 79.9% to 89.5% (Appendix A, Table 1). The yield was not improved further when pH was decreased from 7 to 6.5; therefore, we selected pH 7 as the standard pH value for further optimization. We then tested the temperature effect on this reaction. Despite the fact that the hydrolysis product diminished significantly when the temperature was reduced, the Dha formation was also dramatically decreased (Table 1, Appendix A). At 5 °C, the predominant product was the cyanylated intermediate of Ub-G76C-6H (Table 1, Figure 2D). Another minor side product which we observed when reactions were carried out at room or low temperature was a thionitrobenoate (TNB) adduct of Ub-G76C-6H (Figure 2E). The addition of TCEP was to reduce the disulfide bond, such as in the TNB adduct. Its observation indicated that TCEP was exhausted during the overnight incubation process. TNB is a product of the cyanylation step and will form an oxidized dimer when TCEP is exhausted. This dimer will then react with Ub-G76C-6H to form the TNB adduct. Observations we made so far indicated that the decreased hydrolysis did not result from better selectivity but was due to incomplete reactions. Furthermore, reactions were compared under both denatured and native conditions at 37 °C. To improve the beta elimination process, 10 mM pyridine (Py) was also included in the reaction as a non-nucleophilic base. The results showed a better yield in the presence of 10 mM Py under both denatured (6 M guanidine hydrochloride (GndCl)) and native conditions compared to the same reaction conditions without the addition of 10 mM Py (Table 1), confirming that Py improves the beta elimination process. We also observed slightly more Dha product formed under denaturing conditions than under native conditions, possibly because cysteine was less restrained under denatured conditions (Table 1, Appendix A). It is known that the use of organic solvents will curb hydrolysis. For this reason, we explored how solvent composition influences Dha formation. We made solvent mixtures with different ratios of DMSO to water and performed the Dha formation in the presence of 5 mM NTCB, 0.5 mM TCEP, and 10 mM Py in these solvent mixtures at pH 7 and 37 °C overnight. ESI-MS analysis of reaction products showed that the DMSO percentage increase was correlated with the hydrolysis product decrease (Appendix A), as we expected. However, the addition of DMSO led to a significant amount of the cyanylated Ub-G76C-6H intermediate not being converted into the Dha-containing product (Appendix A). This was contrary to the observation under pure aqueous conditions that typically led to complete conversion of the intermediate. The actual yield of the Dha production formation in the presence of DMSO kept decreasing when the DMSO percentage increased (Table 1). Moreover, a high DMSO percentage also resulted in low protein solubility which made the reaction difficult to perform. Collectively, our results confirmed that the addition of an organic solvent does not improve NTCB-triggered conversion from cysteine to Dha in a protein. Therefore, we concluded that using denatured conditions at pH 7, as well as providing 10 mM pyridine, 0.5 mM TCEP, 5 mM NTCB for a reaction under 37 °C overnight, will provide an effective approach to convert the protein cysteine to Dha.

In eukaryotic cells, proteins can be post-translationally modified by Ub or Ubl proteins to undergo proteasome degradation or serve as function regulators for a large variety of cellular processes [35,36,37,38,39]. To study pathways involving Ub and Ubls, diverse Ub and Ubl probes have been synthesized and used for various research purposes [3,40,41,42]. Dha probes which contain an α, β-unsaturated carbonyl structure at the *C*-terminus of Ub and Ubl proteins are ideal in the formation of covalent adducts with a catalytic cysteine in many enzymes in Ub and Ubl pathways. Mono-Ub, di-Ub, SUMO2, LC3 and NEDD8 based Dha probes have all been successfully synthesized by either intein-based protein semi-synthesis or total synthesis [17,20,43,44,45,46,47]. To simplify the synthesis of these Ub/Ubl-Dha probes, we explored the use of NTCB-triggered Dha formation from cysteine for their generation. We expressed a series of recombinant *N*-terminal FLAG-tagged and *C*-terminal 6×His-tagged Ubl proteins (FLAG-Ubl-GxC-6H), including NEDD8, MNSFβ, GABARAP, GABARAPL2, UFM1, URM1, ISG15 and SUMO1/2/3/4 (Figure 3A). Therein, SUMO1/2/3/4, ISG15 and MNSFβ natively contain a cysteine residue in their sequences. To avoid unexpected modification at these cysteines, they were mutated to alanine or serine. ESI-MS analysis of purified proteins showed their expected molecular weights (Figure 3B). These proteins were then used to undergo the conversion of the installed cysteine at the *C*-terminal glycine position to Dha using the aforementioned optimized conditions. All synthesized Dha-containing products (FLAG-Ubl-GxDha-6H) displayed the expected molecular weights (Figure 3C). The only type of byproducts that we detected was a hydrolysis species (Appendix A) in which the *C*-terminal 6×His tag was cleaved; therefore, all Dha products were easily accumulated and purified by Ni-NTA resin after reactions. In comparison to the intein-based approaches and protein total synthesis, our method is much simpler.

Thus far, all synthesized Dha-containing proteins had Dha at their flexible *C*-terminal ends. To explore whether our NTCB-triggered Dha formation from cysteine would also work with a cysteine in an internal region of a protein, we mutated seven lysine residues (K6, K11, K27, K29, K33, K48, and K63) separately in Ub, expressed the afforded seven Ub mutants in *E. coli*, and then used them to undergo NTCB-triggered Dha formation. All seven expressed proteins had molecular weights matching perfectly with their theoretical values (Figure 4A and Appendix A). However, reaction products varied accordingly for different cysteine mutants. Surprisingly, ESI-LC-MS analysis showed almost complete hydrolysis, with only a trace amount of Dha formation for the reaction between Ub-K63C and NTCB (Figure 4B, and Appendix A). For the other six mutants, the detected predominant species had a mass of 8564.6 Da that matched with their corresponding cyanylated intermediates (Figure 4C and Appendix A). The reaction seemed to be stalled after the cyanylation process. When we switched to using a native condition, the Dha product was still not visible. The only significantly increased peak was for the TNB adduct (Figure 4D and Appendix A), which is likely due to more oxidized TNB dimer formed in a native condition. The reaction was also not obviously improved when we increased the incubating time or temperature (data not shown). Ub is a highly stable protein with a melting temperature higher than 100 °C; therefore, it is likely that all seven mutants do not fully denature in 6 M GndCl to certain extents. Local structural constraints of the introduced cysteine mutations might have prevented the beta elimination process. More investigations are needed to conclude that NTCB-triggered Dha formation does not work for an internal cysteine in a protein. Nevertheless, our current data support that NTCB-triggered Dha formation from cysteine is highly efficient when the cysteine is placed at a structurally flexible region of a protein.

## 3. Materials and Methods

### 3.1. General Materials

Isopropyl β-d-1-thiogalactopyranoside (IPTG) was bought from INDOFINE Chemical Company Inc. (Hillsborough, NJ, USA). Sodium phosphate monobasic, sodium chloride, imidazole, Tris, HEPES and DMSO were all provided by VWR (Radnor, PA, USA). NTCB was purchased from TCI America (Portland, OR, USA). TCEP was obtained from Alfa Aesar (Tewksbury, MA, USA). LC/MS-grade water and acetonitrile for liquid chromatography were both purchased from Fisher Scientific (Waltham, MA, USA) and supplied with 0.1% Optima™ LC/MS-grade formic acid (Fisher Scientific, Waltham, MA, USA). All oligonucleotide primers and DNA sequences used for DNA mutagenesis were synthesized by Integrated DNA Technologies (Coralville, IA, USA). Sanger sequencing was performed by Eton Bioscience Inc. (San Diego, CA, USA).

### 3.2. Plasmid Construction

The expression vectors containing genes coding Ub-G76C-6H and Ub-K6/11/27/29/33/48/63C were generated by site-directed mutagenesis using pETDuet-1-wild-type Ub as a template [31]. The DNA fragments for all Ubls that contained both an *N*-terminal FLAG tag and a *C*-terminal 6×His tag were ordered from Integrated DNA Technologies (Coralville, IA, USA) and cloned into pETDuet-1 vector by double digestion and ligation using NdeI and KpnI-HF (New England Biolabs: Ipswich, MA, USA) to afford expression vectors for them. Native cysteines were converted to serine or alanine using site-directed mutagenesis. All amino acid and DNA sequence information of proteins that are described in this article can be found in the supporting information.

### 3.3. Recombinant Protein Expression and Purification

The expression of all Ub mutants and FLAG-Ubl proteins were performed according to the protocol described previously [23]. Briefly, overnight culture of each protein was inoculated into 1 L 2xYT medium and allowed to grow at 37 °C until OD_600_ reached 0.6–1.0. Then, 1 mM IPTG was added to induce the expression at 18 °C overnight. Upon saturation, cells were collected by centrifugation (6000 rpm, 20 min, 4 °C) and stored at −80 °C if not lysed directly.

For the purification of seven Ub K to C mutants, the bacterial pellet was resuspended in a Ub lysis buffer (50 mM Tris, 1 mM TCEP, pH 7.8) supplied with 0.2 mg/mL lysozyme (Sigma-Aldrich: St. Louis, MO, USA), and lysed by sonication on ice. The total cell lysate was clarified by centrifugation (10,000 rpm, 30 min, 4 °C) and supernatant was collected. After that, 6 M HCl solution was gradually added into the supernatant with constant stirring to adjust the pH value to 1.5–2. The white precipitate was then removed by centrifugation (10,000 rpm, 30 min, 4 °C) and the pH of the acid purification supernatant was adjusted back to 7.8 using 6 M NaOH and then concentrated using an Amicon stirring filtration system with a 5k MWCO membrane (EMD Millipore, Burlington, MA, USA). Subsequently, each protein was desalted into 50 mM ammonium bicarbonate (ABC) buffer by a HiPrep 26/10 Desalting column (GE Healthcare: Chicago, IL, USA) using an NGC™ chromatography system (Bio-Rad Laboratories: Hercules, CA, USA) and analyzed by 15% SDS-PAGE. The concentration of protein solution was measured by the Pierce™ 660 nm Protein Assay Reagent (Thermo Fisher Scientific: Waltham, MA, USA) and dispensed into 100 nmol aliquots for lyophilization. Eventually, protein pellets were either used for chemical reactions or kept at −80 °C for long-term storage.

For the purification of Ub-G76C-6H, Ni binding buffer (50 mM NaH_2_PO_4_, 500 mM NaCl, 5 mM imidazole, 1 mM TCEP, pH 7.8) was used instead of the Ub lysis buffer to resuspend and lyse the cell pellet. The subsequent steps are identical to the purification of Ub-KxC until finishing the acid purification. Acid supernatant was directly passed through high-affinity Ni-charged resin (Genescript: Piscataway, NJ, USA) without concentration. The resin was then washed with a Ni washing buffer (50 mM NaH_2_PO_4_, 500 mM NaCl, 25 mM imidazole, 1 mM TCEP, pH 7.8), and eluted in a 7 mL Ni elution buffer (50 mM NaH_2_PO_4_, 500 mM NaCl, 300 mM imidazole, 1 mM TCEP, pH 7.8). The following desalting, SDS-PAGE, concentration measurement, lyophilization, and storage were as described above.

The purification of all eleven FLAG-Ubl-GxC-6H proteins, except the involvement of the acid purification part, was performed using the same process as that for Ub-G76C-6H.

### 3.4. NTCB-Triggered Dha Formation

The stock solutions of 500 mM TCEP and 500 mM NTCB were prepared in water and DMSO, respectively. For all the optimization reactions, each aliquot of 100 nmol Ub-G76C-6H was dissolved in a buffer mentioned in Table 1, followed by adding 0.5 mM TCEP and 5 mM NTCB sequentially. The reactions were then incubated at 5/18/37 °C for 18 h based on the desired testing requirements. Later, each reaction mixture was desalted to a 50 mM ABC buffer using a HiTrap Desalting column (GE Healthcare: Chicago, IL, USA) in the NGC chromatography system to quench the reaction. Fractions corresponding to the peak UV signal without conductivity changes were collected for ESI-LC-MS analysis.

For reactions on eleven FLAG-Ubl-G76C-6H proteins, instead of changing buffer and temperature for each condition, a buffer containing 20 mM HEPES, 10 mM pyridine at pH 7, as well as 37 °C incubation was used constantly for all Ubl proteins. All other conditions were identical, as mentioned above.

For reactions on seven Ub K to C mutants, besides one set of reactions which was set up using the same conditions as for FLAG-Ubl proteins, another series of reactions using denatured conditions were performed. For that set, the reaction buffer was further supplied with 6 M GndCl (VWR: Radnor, PA, USA) as a denaturant. All other conditions were identical, as mentioned above.

### 3.5. LC-ESI-MS Analysis

Samples for MS were placed in tube inserts inside HPLC sample vials (Agilent Technologies: Santa Clara, CA, USA) for auto injection. The positive LC-ESI-MS was carried out using a Q Exactive Orbitrap mass spectrometer (Thermo Fisher Scientific, Waltham, MA, USA) connected to a liquid chromatography instrument. The settings for all ESI-MS data acquisitions are listed in Table 2. For the liquid chromatography, 100% water and 100% acetonitrile, both supplied with 0.1% formic acid, were used as mobile phases A and B, respectively. All protein samples were separated on an Accucore™ 150-C4 analytical HPLC column (150 mm × 2.1 mm, 2.6 μm particle size) (Thermo Fisher Scientific: Waltham, MA, USA). The scan range of the mass spectrometer and the gradient of liquid chromatography applied for each analysis varied according to Table 3.

### 3.6. Mass Spectra Data Processing

ESI-MS raw data were exported as .txt files using the Xcalibur Qual Browser (Version: 4.1.31.9; Thermo Fisher Scientific: Waltham, MA, USA). Deconvolution of each raw file was processed by Bayesian Protein Reconstruction using Analyst^TM^ QS software (Version: 1.1; Applied Biosystems: Foster City, CA, USA). The deconvolution parameters were set as follows: Adduct: hydrogen; Step mass: 0.1 Da; S/N threshold: 20; Minimum intensity (%): 5; Iteration: 20. Start and stop mass were inputted according to the observed mass data of each protein sample. Deconvoluted results were then replotted using GraphPad Prism (Version: 8.0.2; GraphPad: San Diego, CA, USA) to obtain normalized spectra. After that, normalized deconvoluted results were integrated by a Python script to obtain their experimental averaged mass values. The theoretical monoisotopic peaks distribution and values were calculated using another Python script. All integrated and theoretical data were plotted using GraphPad Prism.

### 3.7. Product Yield Quantitation

Quantitation of the reaction yield was calculated using the peak areas on the mass intensity chromatogram using the Xcalibur Qual Browser (Version: 4.1.31.9; Thermo Fisher Scientific: Waltham, MA, USA). For example, the yield of Dha product (*Y_Dha_%*) was calculated using Equation (1), where *A* refers to each peak area. Accounting for possible tailing effects and background noise, each peak area only included the space above the line between its beginning and ending. In addition, for peaks that showed up on the previous peak’s tailing area, we assumed such contributions were all caused by the detection of new species. Further correction was applied when a single peak contained both Dha product and cyanylation product (where the intensity of the lower signal species was higher than 5% of the intensity of higher signal species). In this case, the ratio of Dha product (*R_Dha_*) was quantified as the proportion of its signal intensity (*I_Dha_*) over the overall integrated deconvoluted mass spectra of both Dha and cyanylation products using Equation (2). The substitution of *A_Dha_*, the numerator in Equation (1), by *A_Dha_* × *R_Dha_* led to Equation (3) for calculating the yield of the Dha product.
(1)$$Y_{Dha}\%=\frac{A_{Dha}}{A_{Dha}+A_{hydro}+A_{TNB}}\times 100\%$$
(2)$$R_{Dha}=\frac{I_{Dha}}{I_{Dha}+I_{CN}}$$
(3)$$Y_{Dha}\%=\frac{A_{Dha}\times R_{Dha}}{A_{Dha}+A_{hydro}+A_{TNB}}\times 100\%$$

## 4. Conclusions

NTCB is a protein-cyanylating reagent which we have used previously for the development of ACPL, a protein ligation technique. By reprograming NTCB for the generation of Dha from cysteine, we optimized the reaction condition in the presence of a denaturant and a non-nucleophilic base at pH 7. Using a Ub with a cysteine at its flexible *C*-terminal side, we were able to obtain a Dha-containing product with a yield of close to completion. Applying this same condition to a number of Ubl proteins with a cysteine installed at their flexible *C*-terminal sides also resulted in their corresponding Dha-containing derivatives. Ub/Ubl-Dha probes are useful in proteomic analyses of cysteine-containing enzymes functioning in Ub and Ubl pathways. Our method dramatically simplifies their synthesis. Its broad adoption for the study of Ub and Ubl pathways is expected. The majority of histone PTMs are found at the flexible *N*-terminal tail of all four histones. The flexible nature of the histone tails potentiates NTCB-triggered Dha formation from cysteine; therefore, we foresee potential applications of our method in the generation of histones with PTM analogues as well for their functional annotation.

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
