# Peer review of "Site-Specific Conversion of Cysteine in a Protein to Dehydroalanine Using 2-Nitro-5-thiocyanatobenzoic Acid"

_molecules, 2021, doi:10.3390/molecules26092619_

Round 1

Reviewer 1 Report

Present manuscript deals with exploration of NTCB reagent for selectively converting Cysteine amino acid of a protein to dehydroalanine. Authors have used ubiquitin protein as an example to study and optimize this transformation. Although, this method is interesting and can be used for a probe development/PTM analogue synthesis, still it requires major optimization for its selectivity and efficiency.

In literature, NTCB reagent is well documented and has been used for demonstrating similar findings. It has been shown earlier (Ref 32-34) that this reagent can be used/leads to the formation of dehydroalanine as well as cleavages peptides at Cysteine residues. Also, this reagent works well for C-terminal cysteine. Hence, I believe this manuscript requires in depth optimization. Authors should provide emphasizes on novelty in their finding and how it can be generalized for its future applications.

With current experimental details, I believe either this manuscript lacks novelty or requires further detailed systematic optimization of the method described in it.

Overall based on significance of research content and scientific soundness, I think this manuscript can be reconsidered for the publication with appropriate justification for above comments.

Reviewer 2 Report

In this manuscript, the authors have shown site-specific conversion of cysteine in a protein to dehydroalanine using 2-nitro-5-thiocyanatobenzoic acid. The authors also showed how the conjugation could be happening and characterized intermediates as well as side products nicely. I would recommend for the publication after the author addresses the following comments.

  • In the introduction section, please show a general scheme for what is known in the literature using NTCB and the presented work i.e. how it is different than the previous work.
  • I feel like the introduction needs some more work.
  • Similarly, in the Results sections, the first paragraph, show the scheme for the NTCB reaction mechanism.
  • The manuscript is full of abbreviations – I would suggest reducing over usage of abbreviations.
  • Figure 1A, write Peak 1 and Peak 2 instead of only 1 and 2
  • Write the m/z for conjugates in Fig 2A
